# OPTIMIZING THE TRADE-OFF BETWEEN UTILITY AND PERFORMANCE IN INTERPRETABLE SLEEP CLASSIFICATION

## ABSTRACT

Deep learning has made significant strides in numerous fields, yet its adoption in healthcare has been slow due to the considerable risks associated with clinical applications. Explainable models are essential to foster trust and accountability. This work examines the trade-off between interpretable techniques for automating sleep state annotation, a critical step in diagnosing sleep disorders. We introduce an interpretable approach, `NormIntSleep`, that produces explanations grounded in clinical guidelines by combining meaningful features with deep neural network embeddings. Furthermore, we propose the metric $Alignment_{DT}$ to quantify domain-grounded interpretability and the resulting utility of explanations. Crucially, `NormIntSleep` outperforms prior interpretable techniques with 0.814–0.847 accuracy, 0.787–0.793 F1-score, 0.759–0.788 $\kappa$, and the hightest $Alignment_{DT}$ score. `NormIntSleep` represents a potentially generalizable interpretable machine learning approach where domain knowledge is essential for safe and efficient implementation in healthcare.

## 1 INTRODUCTION

The widespread adoption of electronic health record (EHR) by healthcare providers has led to a surge in the availability of patient data (Jianxun et al., 2021), paving the way for the development of increasingly capable deep-learning models (Esteva et al., 2019). However, the limited interpretability of these models hinders their widespread use, as comprehending the rationale behind each classification is crucial for mitigating noise and bias (Stiglic et al., 2020). Linear models combined with a robust feature set can offer a certain level of interpretability (Van Der Donckt et al., 2022). However, designing suitable features can be challenging, and incorporating complex features may result in clinically irrelevant interpretations. In this paper, we explore the trade-off between clinical relevance and model performance in the context of human Electroencephalogram (EEG) sleep stage classification and propose a model that seeks to achieve the ideal balance.

In sleep medicine, there is an urgent need to leverage the capabilities of deep learning to help the 70 million adults suffering from sleep disorders such as sleep apnea, insomnia, and narcolepsy (Holder & Narula, 2022). Brain activity must first be classified into sleep-wake states such as wake, rapid eye movement (REM), and non-REM (NREM) sleep to diagnose and treat sleep disorders. The gold standard for sleep classification involves the manual staging of polysomnogram (PSG) by clinicians. However, this method is labor intensive, costly, and prone to human error (Zhang et al., 2022).

Deep learning models have been successfully developed for automated sleep state classification tasks (Lipton, 2016; Perslev et al., 2019; Yang et al., 2021; Dong et al., 2018; Phan et al., 2019; Supratak et al., 2017; Li et al., 2022; Jia et al., 2020; Qu et al., 2020; Phan et al., 2022). However, these models lack interpretability and have been slow to be adopted in the clinical setting due to distrust (Van Der Donckt et al., 2022; Al-Hussaini et al., 2019). In contrast, the American Academy of Sleep Medicine (AASM) sleep scoring manual guidelines (Berry et al., 2012) classifies sleep based on the occurrence of discrete, interpretable neuronal events such as eye movements, oscillatory rhythms, spindles, K-complexes, and slow waves (Berry et al., 2012).

To reconcile the divide between the opaque deep learning models and human-based classification, a recent study presented a feature-based linear model that showed performance on par with deep

neural networks (Van Der Donckt et al., 2022). However, its features were not explicitly designed to align with AASM clinical guidelines. In response, we propose `NormIntSleep` that effectively combines clinically relevant explanations with the high sleep state classification accuracy typically associated with deep learning models. The contributions of this paper are as follows:

- We introduce `NormIntSleep`, a representation learning framework designed to transform deep learning embeddings into a domain-grounded interpretable feature space compatible with glass-box models like decision trees.
- We propose a new metric, $Alignment_{DT}$, for quantifying and thus ensuring the model's alignment with domain-specific knowledge through a decision tree.
- Comprehensive evaluation of `NormIntSleep` is performed using two public sleep classification datasets, benchmarked against state-of-the-art interpretable and deep learning methods.
- `NormIntSleep` combined with a decision tree perfectly aligns with clinical domain knowledge, achieving an $Alignment_{DT}$ score of 1.0 in contrast to the second best score of 0.44.
- `NormIntSleep` outperforms other approaches that aim for clinically relevant interpretations.
- Guidelines are provided for adoption of `NormIntSleep` in other applications.

## 2  DATA

Table 1: Datasets

| Dataset | Number of Subjects | Sampling Frequency (Hz) | Channel Names | Annotation Schema |
|---|---|---|---|---|
| ISRUC (Khalighi et al., 2016b) | 100 | 200 | F3-A2, C3-A2, F4-A1, C4-A1, O1-A2, O2-A1, ROC-A1, LOC-A2, Chin-EMG | AASM |
| PhysioNet (Kemp et al., 2000) | 197 | 100 | EEG Fpz-Cz, EEG Pz-Oz, EOG horizontal, EMG submental | R&K |

We evaluated sleep staging interpretability and performance using two public datasets (Table 1):

- The ISRUC-SLEEP Dataset (ISRUC) (Khalighi et al., 2016a) includes PSG recordings of 100 human subjects, some diagnosed with sleep apnea and some on medications.
- The Sleep-EDF Database Expanded (PhysioNet) (Kemp et al., 2000; Goldberger et al., 2000) comprises 197 human subjects divided into two cohorts, the first in healthy controls and the second with insomnia receiving a sleep aid medication (temazepam). We combined both cohorts in our experiments to assess generalization capability. Sleep stages N3 and N4, annotated using the R&K schema (, JSSR; Rechtschaffen, 1968), were merged into a single N3 class to conform to AASM standards (Berry et al., 2012; Moser et al., 2009; Danker-hopfe et al., 2009).

## 3  NORMINTSLEEP METHOD

The objective of `NormIntSleep` is to offer a modular framework that can be adapted to various domains, wherein domain knowledge can guide the desired interpretation. We demonstrate the effectiveness of this approach for automatic sleep state classification.

Table 2 defines the notations used. The input consists of multi-channel physiological signals (EEG, EOG, EMG) divided into 30 sec segments called epochs. During clinical annotation, the annotator assigns a sleep stage label to each 30 sec segment by examining the signals. There are five possible sleep stages: Wake (W), Rapid Eye Movement (REM), Non-REM 1 (N1), N2, and N3. The goal of `NormIntSleep` is to predict these sleep stages for each epoch ($y_i \in$ W, N1, N2, N3, REM) based on the physiological signals ($\boldsymbol{x}_i \in \mathbb{R}^{C \times fs \cdot 30}$) while providing a meaningful interpretation.

`NormIntSleep` architecture is detailed in the following sections and illustrated in Figure 1.

### 3.1  PRE-TRAINING

`NormIntSleep` utilizes the PSG recordings, $\mathcal{X}$, to generate an interpretable representation for deep neural network embeddings. The pre-training algorithm for creating the linear projector is outlined in Algorithm 1 and depicted in Figure 1a.

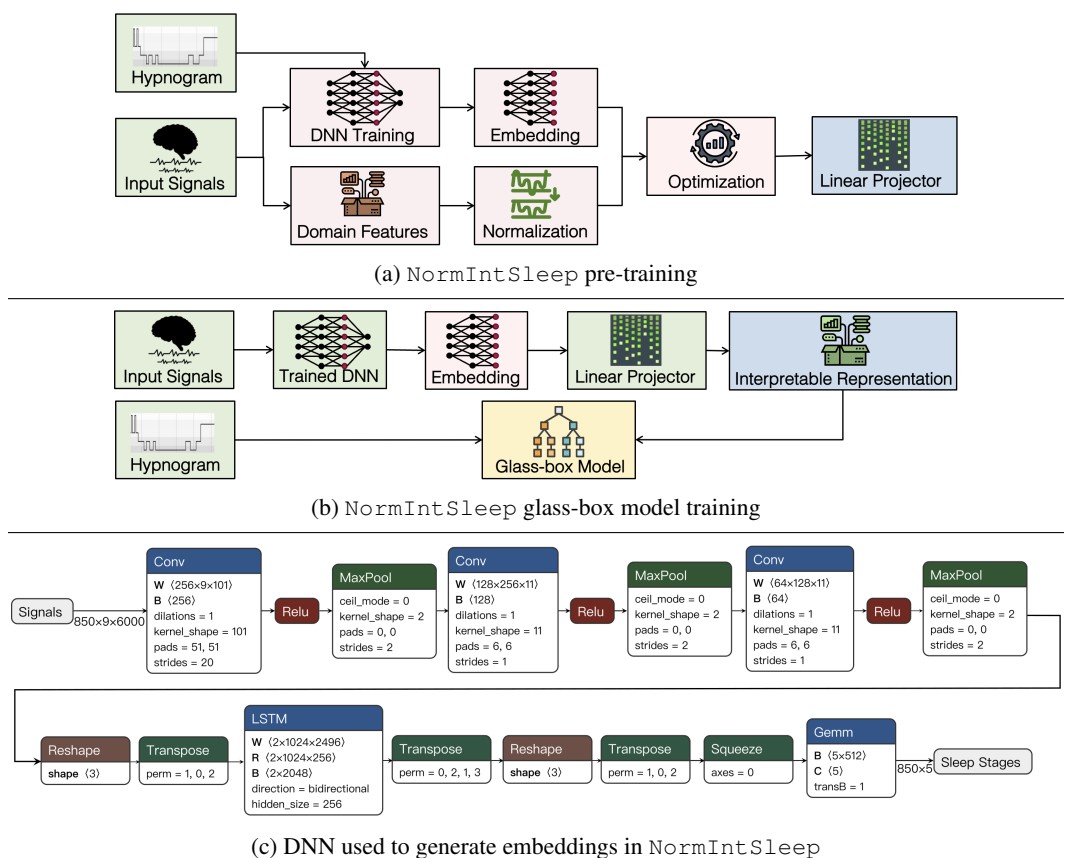

(a) `NormIntSleep` pre-training

(b) `NormIntSleep` glass-box model training

(c) DNN used to generate embeddings in `NormIntSleep`

Figure 1: Architecture of `NormIntSleep`. (a) The pre-training schema where DNN training and subsequent embedding generation occurs in parallel to domain feature extraction. The results are combined to produce the linear projector. (b) The glass-box model is trained using the interpretable representations of the DNN embeddings generated using the linear projector. (c) The architecture of the DNN used in `NormIntSleep`.

A Deep Neural Network (DNN) is trained end-to-end on sleep staging. The multi-channel EEG, EOG, and EMG signals ($\mathcal{X}$) are used as input. The CNN consists of three convolutional layers, with each layer followed by batch normalization, ReLU activation, and max pooling. The kernel sizes of the three layers are 201, 11, and 11, and the output channels are 256, 128, and 64. The CNN output serves as input for a

Table 2: Notations

| Symbol | Meaning |
|---|---|
| N | Number of subjects |
| M | Number of epochs |
| C | Number of channels |
| fs | Sampling frequency |
| $\boldsymbol{X_j} \in \mathcal{X}$ | Input signals for subject $j$ |
| $\boldsymbol{Y_j} \in \mathcal{Y}$ | Sleep stages for subject $j$ |
| $\boldsymbol{x}_i \in \mathbb{R}^{C \times fs \cdot 30}$ | Input signals in an epoch |
| $\boldsymbol{y}_i \in \{$W, N1, N2, N3, REM$\}$ | Sleep stage in an epoch |
| $\boldsymbol{F}(\boldsymbol{x}_i) \in \mathcal{F}$ | Interpretable features of epoch $\boldsymbol{x}_i$ |
| $\boldsymbol{E}(\boldsymbol{x}_i) \in \mathcal{E}$ | Embeddings of epoch $\boldsymbol{x}_i$ |
| $\boldsymbol{T}$ | Linear Projector |
| $\boldsymbol{R}(\boldsymbol{x}_i) \in \mathcal{F}_N$ | Interpretable representation of $\boldsymbol{E}(\boldsymbol{x}_i)$ |
| $\boldsymbol{h}$ | Glass-box Model |

layer of bi-directional Long Short-Term Memory (LSTM) cells with 256 hidden states. The resulting 512 hidden states constitute the *embedding space*, $\mathcal{E}$. During model training with cross-entropy loss, the LSTM output, $\boldsymbol{E}(\boldsymbol{X})$, connects to a fully-connected layer featuring five outputs corresponding to the five sleep stages. The network is illustrated in Figure 1c.

---

**Algorithm 1: `NormIntSleep` pre-training**

---

**Input:** training data: $\{\mathcal{X}, \mathcal{Y}\}$

**Output:** linear projector: $\boldsymbol{T}$

1: **for** $\{\boldsymbol{X_j}, \boldsymbol{Y_j}\} \in \{\mathcal{X}, \mathcal{Y}\}$ **do**      // ** *training dataset* **//

2: $\quad \boldsymbol{F_N}(\boldsymbol{X_j}) \leftarrow \dfrac{\boldsymbol{F}(\boldsymbol{X_j}) - \boldsymbol{\mu_F}}{\boldsymbol{\sigma_F}}$      // ** *normalization* **//

3: $\quad$ **for** $\{\boldsymbol{x_i}, \boldsymbol{y_i}\} \in \{\boldsymbol{X_j}, \boldsymbol{Y_j}\}$ **do**      // ** *train DNN* **//

4: $\qquad \boldsymbol{E}(\boldsymbol{x_j}) \leftarrow$ Embeddings      // ** *output of (n-1)-th layer in n-layer DNN* **//

5: $\qquad \boldsymbol{z_i} \leftarrow \boldsymbol{W}^T \boldsymbol{E}(\boldsymbol{x_i}) + \boldsymbol{b}$      // ** *fully-connected layer Layer* **//

6: $\qquad \sigma(s_i[c]) = \dfrac{e^{z_i[c]}}{\sum\limits_{k=1}^{5} e^{z_i[k]}}$      // ** *softmax* **//

7: $\qquad L(\boldsymbol{y_i}, \boldsymbol{s_i}) = -\sum\limits_{k}^{5} \boldsymbol{y_i}[k] log(\sigma(s_i[k]))$      // ** *loss* **//

8: $\quad$ **end for**

9: **end for**

10: $\boldsymbol{F}(\mathcal{X}) \leftarrow$ features      // ** *domain guided features* **//

11: $\boldsymbol{\mu_F} \leftarrow$ means of $\boldsymbol{F}(\mathcal{X})$      // ** *for each column, i.e. feature* **//

12: $\boldsymbol{\sigma_F} \leftarrow$ standard deviation of $\boldsymbol{F}(\mathcal{X})$      // ** *for each column, i.e. feature* **//

13: $\boldsymbol{F_N}(\mathcal{X}) \leftarrow \dfrac{\boldsymbol{F}(\mathcal{X}) - \boldsymbol{\mu_F}}{\boldsymbol{\sigma_F}}$      // ** *normalization of features* **//

14: $\boldsymbol{T}' \leftarrow \min\limits_{T'} ||\boldsymbol{E}(\mathcal{X}) - \boldsymbol{F_N}(\mathcal{X})\boldsymbol{T}'||_2^2$      // ** *least squares optimization* **//

---

Features $\boldsymbol{F}(\boldsymbol{X})$, defined in the *feature space* $\mathcal{F}$, are extracted from the dataset. These features, selected based on clinical guidelines, are described in detail in Section 3.3. Standardization is used to normalize the features as follows: $\boldsymbol{F_N}(\boldsymbol{X}) \leftarrow \dfrac{\boldsymbol{F}(\boldsymbol{X}) - \boldsymbol{\mu_F}}{\boldsymbol{\sigma_F}}$

A linear transformation, $\boldsymbol{T}$, is learned to map the embedding space to the normalized feature space, $\mathcal{E} \xrightarrow{\boldsymbol{T}} \mathcal{F}_N$, using least squares regression. $\boldsymbol{R}(\boldsymbol{X}) = \boldsymbol{E}(\boldsymbol{X}) \cdot \boldsymbol{T}$ defines the interpretable representations obtained after projecting the embedding, $\boldsymbol{E}(\boldsymbol{X})$, to the normalized feature space, $\mathcal{F}_N$.

**Ablation Studies.** The two primary components of `NormIntSleep` that influence its performance are studied in Appendix D: (1) *Normalization scheme* alters the distribution of both the feature and interpretable representations of the embeddings, thereby impacting utility and overall performance. (2) *Method for learning linear projector* governs the efficacy of the projection matrix used to map the embeddings onto the feature space, consequently affecting the performance of the model.

## 3.2 GLASS-BOX MODEL TRAINING

The procedures for generating interpretable representations from the embeddings, training the glass-box model, and conducting the inference process to procure both predictions and explanations are laid out in Algorithm 2. Figure 1b shows a visual representation of this process.

Initially, the embeddings are created using the pre-trained DNN. The linear projection matrix, $\boldsymbol{T}$, is subsequently employed to project the embeddings onto the interpretable feature space, resulting in $\boldsymbol{R}(\boldsymbol{X}) \leftarrow \boldsymbol{E}(\boldsymbol{X}) \cdot \boldsymbol{T}$. These representations of the embeddings reside in the normalized interpretable feature space, $\mathcal{F}_N$. Finally, these interpretable representations, along with the hypnogram, are used to train a glass-box model, $\boldsymbol{h}$.

During inference, a similar sequence is followed. The embeddings are first generated using the pre-trained DNN. The linear projection matrix, $\boldsymbol{T}$, is then applied to project the embeddings onto the interpretable feature space. The resulting interpretable representations are input into the trained glass-box model to obtain the predicted sleep stages.

---

**Algorithm 2: `NormIntSleep` glass-box model training and inference**

---

**Input:** training data: $\{\mathcal{X}\mathcal{Y}\}$; test data: $\{\mathcal{X}'\}$; linear projector: $\boldsymbol{T}$; pre-trained DNN with last layer removed: $\boldsymbol{E}$

**Output:** predicted classes and explanations: $\{\mathcal{Y}', \mathcal{Z}'\}$

    **# Training**

1: **for** $X_j \in \mathcal{X}$ **do**                                                            *// \*\* training dataset \*\*//*

2:     $E(X_j) \leftarrow$ Embeddings

3:     $R(X_j) \leftarrow E(X_j) \cdot T$     *// \*\* interpretable representations of embeddings \*\*//*

4: **end for**

5: Train glass-box model, $\boldsymbol{h}$, using $\{\boldsymbol{R}(\mathcal{X}), \mathcal{Y}\}$

    **# Inference**

6: **for** $\{X'_j\} \in \{\mathcal{X}'\}$ **do**                                                       *// \*\* test dataset \*\*//*

7:     $E(X'_j) \leftarrow$ Embeddings

8:     $R(X'_j) \leftarrow E(X'_j) \cdot T$     *// \*\* interpretable representations of embeddings \*\*//*

9:     $\{Y'_j, Z'_j\} \leftarrow h(R'(X'_j))$     *// \*\* Predicted classes and explanations \*\*//*

10: **end for**

---

## 3.3 DOMAIN FEATURES

The linear projector in Figure 1a utilized in `NormIntSleep` learns from a meticulously crafted feature set named *FeatShort*, which is derived from domain-specific knowledge. *FeatShort* consists of clinically relevant features designed based on AASM manual guidelines (Berry et al., 2012), with 121 features for the ISRUC dataset and 52 for the Physionet dataset, some of which are adapted from (Al-Hussaini & Mitchell, 2022). *FeatShort* extracts the following features:

- **Complexity, Mobility**: Complexity measures the similarity of the signals to a pure sine wave, converging when the frequency is constant. N3 is usually characterized by constant, low-frequency waves, and REM by complex, high-frequency activity. Mobility represents the mean frequency of the signal. These features were extracted using AntroPy.
- **Rapid eye movements** were only extracted for the ISRUC dataset as it requires two EOG channels. The approaches proposed by Yetton et al. (2016); Agarwal et al. (2005) are used by YASA (Vallat & Walker, 2021) to extract REM. It is the primary characteristic of the REM sleep stage.
- **Delta, Theta, Alpha, Beta** band powers: Delta (0.5-4Hz) waves are used to annotate N3, Theta (4-8Hz) waves are used to classify N1, Alpha (8-12Hz) and Beta (>12Hz) waves distinguish N1 from Wake (Berry et al., 2012). In EMG, these bands help differentiate between Wake and REM. Power Spectral Density in each band is estimated using Welch's method (Virtanen et al., 2020).
- **Sleep spindles** is a defining feature of N2 sleep stage. The method proposed by (Lacourse et al., 2019) and (Vallat & Walker, 2021) was used to extract spindles.
- **Slow waves**. The N3 sleep stage is defined by the presence of low-frequency, high-amplitude, and delta activity called slow waves. Methods by (Carrier et al., 2011), (Massimini et al., 2004), and (Vallat & Walker, 2021) were used for extraction.
- **#-Zero-Crossings** represent the number of times the signal oscillates between value extremes. A high number of crossings indicates high fluctuating activity in the corresponding signal. NREM slow waves are usually classified based on the frequency at which zero-crossings occur. This feature was extracted using AntroPy.
- **Amplitude** is a distinguishing feature in many underlying characteristics used for sleep staging like K Complexes and Low Amplitude Mixed Frequency (Berry et al., 2012).
- **Kurtosis, skewness, variance, mean** represent the distribution of epochs and reveal other underlying traits. For example, previous work discovered a relation between variance in EEG and delta waves (Mariani et al., 2011). These features were extracted using SciPy (Virtanen et al., 2020).

In contrast to *FeatShort*, which contains only clinically relevant features, *FeatLong* (Van Der Donckt et al., 2022) presents a more exhaustive feature list (Appendix A), utilized alongside glass-box models without incorporating the `NormIntSleep` architecture. A goal of this study is to juxtapose

the efficacy of `NormIntSleep` in combination with *FeatShort* against the comprehensive but clinically less meaningful *FeatLong*. The *FeatLong* features do not emphasize the clinical guidelines as proposed in the AASM Manual (Berry et al., 2012), yielding 2488 features for the ISRUC dataset and 1048 for the Physionet dataset. Through analysis of variance, the topmost 90% significant features are retained in both *FeatShort* and *FeatLong*.

### 3.4 METRIC FOR DOMAIN-GROUNDED INTERPRETABILITY

`NormIntSleep` aims to align explanations with domain knowledge. This ensures that the insights generated are valuable to clinicians, making our findings more actionable. To quantify this alignment between a decision tree and clinical domain knowledge, we introduce a metric, $Alignment_{DT}$ $\in [0, 1]$. Nodes predominantly consisting of a single sleep stage ($> 95\%$) are disregarded during calculation as they do not play a major role in the overall model behavior of the tree and behave similar to a leaf node. It is defined as follows, with a desired value of 1.0 during perfect alignment:

$$Alignment_{DT} = \frac{\text{Number of nodes that align with clinical domain knowledge}}{\text{Total number of nodes}} \quad (1)$$

## 4 EXPERIMENTS

### 4.1 EXPERIMENTAL SETUP

The DNN was trained using PyTorch (Paszke et al., 2019) with a batch size of 1000 samples from a single PSG. Training continued for 50 epochs with a starting learning rate of $10^{-4}$, using the Adam (Kingma & Ba, 2014) optimization method. Glass-box models were trained using scikit-learn (Pedregosa et al., 2011; Raschka et al., 2020), XGBoost (Chen & Guestrin, 2016), and CatBoost (Prokhorenkova et al., 2018). Features were extracted using scikit-learn, MNE (Gramfort et al., 2013), YASA (Vallat & Walker, 2021), and tsflex (Van Der Donckt et al., 2021).

We partitioned data by subjects into a 9:1 training-test split using the same seed for all experiments. As a result, the same subjects were consistently used for testing in every experiment. Model hyperparameters were optimized based on the training data, while the test data was used to obtain performance metrics. To avoid overfitting and maintain model consistency across datasets, we used identical model hyperparameters and feature extraction methods for both datasets. Comprehensive implementation details, including hyperparameters and baselines, are provided in Appendix B.

### 4.2 BASELINES

We used a variety of existing deep learning models as benchmarks to assess our model:

- U-Time (single-channel) (Perslev et al., 2019): deep learning model based on U-Net architecture.
- DeepSleepNet (single-channel) (Supratak et al., 2017): combines CNN and LSTM.
- CNN: Convolutional Neural Network proposed in (Al-Hussaini et al., 2019).
- TinySleepNet (single-channel) (Supratak & Guo, 2020): CNN and LSTM.
- AttnSleep (single-channel) (Eldele et al., 2021): an attention-based approach.

We also designed and utilized the following deep learning models by either adapting established methodologies or creating new ones (Implementation details: Appendices B and F):

- U-Time (multi-channel): adapted the U-Time architecture to integrate multiple channels to achieve state-of-the-art performance. Detailed architecture illustrated in Figure 8 in the Appendix.
- DeepSleepNet (multi-channel): modified the DeepSleepNet architecture to include multiple channels for improved performance. Detailed architecture can be found in Figure 7 in the Appendix.
- AttentionNet: introduced a deep neural network architecture that consists of convolutional layers and multi-headed attention. Architecture illustrated in Figure 9 in the Appendix.
- RCNN (DNN): the proposed DNN that serves as the foundation of `NormIntSleep` (Figure 1c).
- RCNN-MHA: modified the proposed RCNN architecture to include residual connections and multi-headed attention (MHA). Architecture illustrated in Figure 10 in the Appendix.
- RCNN-SDPA: further modified the proposed RCNN to incorporate residual connections and scaled dot product attention (SDPA). Architecture illustrated in Figure 11 in the Appendix.

Table 3: Comparison of interpretable and black box deep learning methods with `NormIntSleep`. The best interpretable method using `NormIntSleep`, the best feature-based method, and the best black-box methods are bolded.

| | Model | Accuracy | | F1 Score (Macro) | | Cohen's $\kappa$ | |
|---|---|---|---|---|---|---|---|
| | | Physionet | ISRUC | Physionet | ISRUC | Physionet | ISRUC |
| Interpretable Methods | SLEEPER-GradientBoostedTrees (Al-Hussaini et al., 2019) | 0.807 | 0.797 | 0.721 | 0.756 | 0.729 | 0.736 |
| | SERF-XGBoost (Al-Hussaini & Mitchell, 2022) | 0.823 | 0.819 | 0.753 | 0.789 | 0.753 | 0.766 |
| | FeatLong (Van Der Donckt et al., 2022)-XGBoost | 0.861 | 0.809 | 0.810 | 0.775 | 0.809 | 0.752 |
| | **FeatLong (Van Der Donckt et al., 2022)-CatBoost** | **0.862** | **0.811** | **0.811** | **0.775** | **0.810** | **0.754** |
| | FeatLong (Van Der Donckt et al., 2022)-LogisticRegression | 0.856 | 0.800 | 0.801 | 0.762 | 0.801 | 0.741 |
| | FeatShort-XGBoost | 0.828 | 0.791 | 0.768 | 0.752 | 0.764 | 0.728 |
| | `NormIntSleep`-XGBoost | 0.845 | 0.811 | 0.787 | 0.783 | 0.785 | 0.755 |
| | FeatShort-CatBoost | 0.834 | 0.798 | 0.776 | 0.755 | 0.771 | 0.737 |
| | **`NormIntSleep`-CatBoost** | **0.847** | **0.814** | **0.793** | **0.787** | **0.788** | **0.759** |
| | `NormIntSleep`-LogisticRegression | 0.853 | 0.788 | 0.797 | 0.764 | 0.796 | 0.723 |
| | FeatShort-Decision Tree (Depth 7) | 0.758 | 0.698 | 0.683 | 0.591 | 0.663 | 0.596 |
| | `NormIntSleep`-Decision Tree (Depth 7) | 0.819 | 0.791 | 0.751 | 0.761 | 0.749 | 0.728 |
| Deep Learning | U-Time (Perslev et al., 2019) | 0.805 | 0.807 | 0.743 | 0.779 | 0.733 | 0.751 |
| | DeepSleepNet (Supratak et al., 2017) | 0.841 | 0.811 | 0.788 | 0.778 | 0.782 | 0.754 |
| | CNN (Al-Hussaini et al., 2019) | 0.851 | 0.808 | 0.796 | 0.778 | 0.794 | 0.752 |
| | AttnSleep (Eldele et al., 2021) | 0.832 | 0.807 | 0.772 | 0.760 | 0.771 | 0.749 |
| | TinySleepNet (Supratak & Guo, 2020) | 0.811 | 0.796 | 0.731 | 0.751 | 0.739 | 0.735 |
| | **U-Time (multi-channel)** | **0.868** | **0.849** | **0.822** | **0.825** | **0.820** | **0.805** |
| | DeepSleepNet (multi-channel) | 0.867 | 0.822 | 0.820 | 0.803 | 0.817 | 0.768 |
| | AttentionNet | 0.841 | 0.806 | 0.781 | 0.774 | 0.781 | 0.750 |
| | RCNN (DNN in `NormIntSleep`) | 0.863 | 0.844 | 0.817 | 0.822 | 0.811 | 0.799 |
| | RCNN-MHA | 0.857 | 0.811 | 0.811 | 0.788 | 0.805 | 0.757 |
| | RCNN-SDPA | 0.867 | 0.790 | 0.823 | 0.774 | 0.818 | 0.726 |

For model performance evaluation, we also made use of the following interpretable benchmarks:

- *FeatShort* and glass-box models: We combined the features outlined in Section 3.3 with glass-box models, bypassing the use of the `NormIntSleep` architecture.
- *FeatLong* (Van Der Donckt et al., 2022) and glass-box models: We paired the features described in Appendix A with glass-box models.
- SLEEPER (Al-Hussaini et al., 2019): Prototype-based interpretable algorithm for sleep staging.
- SERF (Al-Hussaini & Mitchell, 2022): Interpretable method using embeddings, rules, features.

**Metrics.** Model performance is compared using Accuracy, Macro F1-Score, and Cohen's $\kappa$.

## 5 RESULTS

Table 3 compares the sleep classification performance of `NormIntSleep` with the methods detailed in Section 4.2. The F1-scores for each of the sleep stages are presented in Tables 7 and 6 in the Appendix. We then evaluate the importance of features critical for sleep state classification from a clinical perspective and quantify the alignment with domain knowledge using $Alignment_{DT}$. The top performing interpretable, feature-based method was *FeatLong*-CatBoost with an accuracy of 0.811–0.862, F1 of 0.775–0.811, and Cohen's $\kappa$ of 0.754–0.810. `NormIntSleep`–CatBoost demonstrates an accuracy of 0.814–0.847, F1 of 0.787–0.793 and Cohen's $\kappa$ of 0.759–0.788. The top performing black-box deep learning method was U-Time with an accuracy of 0.849–0.868, F1 of 0.822-0.825, and Cohen's $\kappa$ of 0.805–0.820. The results demonstrate that `NormIntSleep` surpasses the performance of all other interpretable methods, with the sole exception of the exhaustive feature list present in *FeatLong*. Additionally, for the most interpretable glass-box method, Decision Tree, `NormIntSleep` Tree improves the accuracy of the corresponding feature-based approach from 75.8% to 81.9% for the Physionet dataset and 69.8% to 79.1% for the ISRUC dataset, further demonstrating the efficacy of the `NormIntSleep` framework. Results for different demographics and confidence intervals are in Appendices J and I.

### 5.1 INTERPRETABLE DECISION TREE

`NormIntSleep` uses clinical AASM guidelines to generate a list of features (*FeatShort*) that is used as a representation of the DNN embeddings. Figure 2 is the decision tree generated by `NormIntSleep`-DecisionTree showing counts for the entire training set in the PhysioNet dataset.

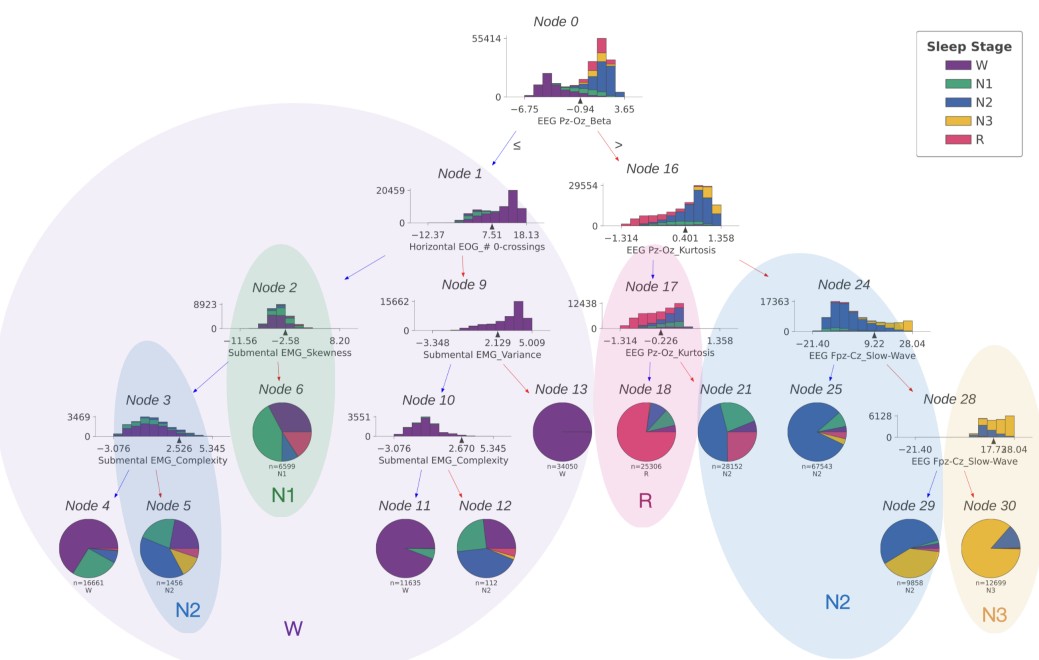

Figure 2: `NormIntSleep`-Decision Tree for PhysioNet. The tree depth has been limited to 4 levels to facilitate examination and provide a clear understanding. The histogram stacks represent tree nodes, and the pie charts symbolize the leaves. Each node displays the representative feature value distribution for the five sleep stages, with a marked upward arrow indicating the threshold for node splitting. This short and pruned decision tree achieves an accuracy of 79.4%, a kappa value of 0.707, and a perfect $Alignment_{DT}$ score of 1.0.

`NormIntSleep`-DecisionTree attained an accuracy of approximately 79%. In comparison, *Feat-Short*-DecisionTree, which utilizes the features as input to the glass-box models, obtained an accuracy of 71% on the same dataset. To judge the interpretability of the decision tree, a practicing sleep clinician made the following observations of the nodes within the decision tree.

- **Node 0** employs beta waves, with frequencies between 8–20 Hz, in EEG, resulting in a significantly higher proportion of Wake on the split with a higher absolute value. In clinical practice, beta waves are dominant during Wake in comparison to the other sleep stage of NREM and REM (Patel et al., 2021; Estrada et al., 2004).
- **Node 1** splits based on the number of EOG crossings, resulting in more Wake on the side with a higher value. In clinical practice, high EOG activity signifies wakefulness (Ganesan & Jain, 2020) and eye movements associated with REM (Lin et al., 2019; Barea et al., 2012; Herman et al., 1984; Boukadoum & Ktonas, 1986).
- **Nodes 2 and 3** utilizes high EMG Complexity and Skewness to differentiate Wake from other sleep stages. In clinical practice, Wake has high EMG activity (both magnitude and variance). Moreover, EMG variance is also high in N1 relative to N2, N3, and REM, thus explaining the increased amount of N1 in subsequent nodes.
- **Nodes 16 and 17** employ low kurtosis in EEG to classify REM. In clinical practice, REM sleep is characterized by homogenous EEG (Krauss et al., 2018), resulting in low kurtosis (platykurtic).
- **Nodes 24 and 28** use slow waves in an EEG channel to differentiate N3 from N2. In clinical practice, presence of slow waves are used to classify N3 (Patel et al., 2021; Berry et al., 2012).

The decision tree underscores the effectiveness of `NormIntSleep` in projecting the embeddings onto an interpretable feature space with clinical relevance. It demonstrates that the paths taken during tree traversal (beta waves, EOG crossings, EMG complexity, EEG kurtosis, and amount of delta waves) mirror clinical decision-making.

## 5.2 QUANTIFICATION OF ALIGNMENT WITH DOMAIN KNOWLEDGE

Based on the explanations in Section 5.1 and Figure 2, `NormIntSleep` achieves an $Alignment_{DT}$ (Eq. 1) score of 1.0. Nodes 9 and 10 due to $> 95\%$ (explanation: Section 3.4) Wake composition. In contrast, the *FeatLong*-DecisionTree (depth = 4) obtains a $Alignment_{DT}$ score of 0.22, excluding nodes 9 and 13, as illustrated in Figure 12 of the Appendix. The $Alignment_{DT}$ score for *SERF* based on the tree presented in paper is 0.44. It's worth noting that *SERF* leverages features that are more rooted in clinical guidelines than *FeatLong*. Hence, `NormIntSleep` offers explanations that are substantially better grounded in domain knowledge compared to other interpretable methods.

## 5.3 FEATURE IMPORTANCE

Using SHapley Additive exPlanations (SHAP) (Lundberg & Lee, 2017), Figure 3 demonstrates the top five most important dimensions of the interpretable representation of the `NormIntSleep` embeddings paired with XG-Boost and their impact on the five sleep stages. Figure 6 in the appendix shows the distribution of SHAP values of the top five interpretable dimensions for individual sleep stages. The top five most important features in Figure 3 were considerably more critical than the rest, so only these five features and their implications are explained below:

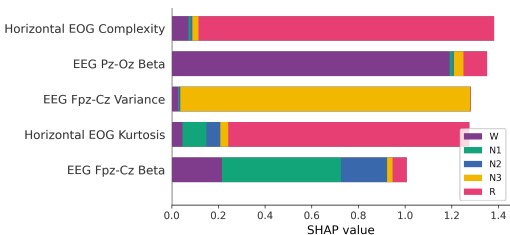

Figure 3: Most important embedding representations in `NormIntSleep`-XGBoost and the influence on each sleep stage classification according to SHAP values

- The first and fourth most important features emphasize the significance of EOG complexity and kurtosis in classifying REM, which is consistent with Figure 2, Node 1, and Figure 6(e). EOG activity is very high during REM (Boukadoum & Ktonas, 1986; Herman et al., 1984).
- The second most important feature underlines the importance of beta waves in classifying Wake and is consistent with Figure 2, Node 0, and Figure 6(d). Beta waves are dominant during wakefulness (Patel et al., 2021; Estrada et al., 2004).
- The third feature indicates the importance of EEG variance in classifying N3. Consistent with Figure 2, Nodes 0, 24, and 28, high EEG variance is a common attribute of the high amplitude and low-frequency slow waves (Frauscher et al., 2015), a characteristic of the N3 sleep stage. This is also consistent with Figure 6(c).
- The fifth feature emphasizes the importance of beta waves in EEG in distinguishing Wake from N1. Beta waves are dominant during wakefulness (Patel et al., 2021; Estrada et al., 2004) but not during N1, making them the perfect attribute to differentiate between the two, as highlighted in Section 5.1, Figure 2, Node 6, and Figure 6(a), 6(d).

The alignment of the SHAP values with clinical guidelines underscores the utility of the explanations provided by `NormIntSleep`. Moreover, as secondary validation, a practicing clinical sleep specialist reviewed the interpretations of `NormIntSleep`. The detailed procedure for adoption of `NormIntSleep` in other domains is detailed in Appendix K.

## 6 DISCUSSION

Interpretability is crucial for the adoption of clinical decision support systems. Complex features paired with a simple model can provide interpretation. However, those explanations can only be helpful if the features are clinically meaningful. This study proposed the integrated use of `NormIntSleep` with *FeatShort*, a feature-set grounded in clinical guidelines from the AASM manual. This combination results in a highly accurate, more clinically interpretable sleep classification. In contrast, while comparably accurate at classifying sleep states, *FeatLong* and *SERF* falls short of full compliance with the AASM manual and therefore has limited practical utility for clinicians. Through the proposed metric $Alignment_{DT}$, we quantify the significantly better alignment of `NormIntSleep` with domain expertise relative to other methods. In conclusion, we demonstrate that interpretability can be successfully implemented in sleep state classification and is a critical step towards wider adoption into clinical decision support systems in all of healthcare.

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

# A   FEATURES IN FEATLONG

In this appendix, we delve into the details of the expansive *FeatLong* feature set, comprising both time-domain and frequency-domain attributes. These features provide an extensive perspective on the signal data. However, it is vital to underscore that while these attributes offer a broad understanding of the signal's properties, some of these features are not in accordance with the AASM Manual. This lack of direct alignment could potentially limit their utility in clinical contexts.

## A.1   TIME-DOMAIN FEATURES

Time-domain features primarily capture the statistical characteristics and the unique waveform properties of the signal within the temporal spectrum:

- **Standard Deviation, Interquartile Range, Skewness, Kurtosis:** These features are part of the more concise *FeatShort* feature set discussed in the main paper, describing the statistical distribution of the signal epochs.
- **Number of zero-crossings:** This feature counts the instances where the signal swings between value extremes, playing a significant role in the categorization of NREM slow waves.
- **Hjorth Mobility, Hjorth Complexity:** These features align with 'Mobility' and 'Complexity' within *FeatShort*, representing the mean frequency of the signal and its resemblance to a pure sine wave, respectively.
- **Higuchi Fractal Dimension, Petrosian Fractal Dimension, Permutation Entropy, Binned Entropy:** These intricate features capture complex structures of the signal, including its fractal dimension and entropy. However, their absence from the AASM Manual may limit their clinical applicability.

## A.2   FREQUENCY-DOMAIN FEATURES

Frequency-domain features shed light on the spectral properties of the signal:

- **Spectral Fourier Statistics, Binned Fourier Entropy:** These features capture the spectral aspects of the signal, such as the statistical properties and entropy of its Fourier transform. However, their absence from the AASM Manual may limit their clinical relevance.
- **Absolute and Relative Spectral Power in the 0.4-30Hz Band:** These measures mirror the Delta, Theta, Alpha, and Beta band powers in the *FeatShort* set, facilitating the classification of different sleep stages.
- **Fast Delta + Theta Spectral Power, Alpha / Theta Spectral Power, Delta / Beta Spectral power, Delta / Sigma Spectral Power, Delta / Theta Spectral Power:** These metrics represent the ratio or aggregate power in specific spectral bands, highlighting the interaction between different frequency components of the signal. While they can provide valuable insights for sleep staging, they are not explicitly outlined in the AASM Manual.

Through this appendix, we have endeavored to shed light on the extensive *FeatLong* feature set, underscoring its depth and range. Nevertheless, we must acknowledge the trade-off between the comprehensive nature of *FeatLong* and the clinically oriented *FeatShort*. This arises due to some features in *FeatLong* not directly complying with the AASM Manual, potentially impacting their clinical relevance. This nuanced distinction between the two feature sets is further explored in the main body of the paper.

# B    IMPLEMENTATION DETAILS

The data was split by patients into training and test set using a 9:1 ratio. The model hyper-parameters were set using the training set, including those for the glass-box models. All neural networks were trained using PyTorch 1.0 (Paszke et al., 2019) using ADAM (Kingma & Ba, 2014) as the optimization method. A batch size of 1000 epochs was used. The training was performed for 50 epochs. The learning rate was reduced by a factor of 10, after 30 epochs. A learning rate of $10^{-4}$ was used. Comprehensive descriptions of the deep learning models proposed in this paper can be found in Appendix F.

Glass-box models were developed using scikit-learn (Pedregosa et al., 2011; Raschka et al., 2020), XGBoost (Chen & Guestrin, 2016), and CatBoost (Prokhorenkova et al., 2018). Feature extraction was performed using methods discussed in Section 3.3 of the main manuscript, as well as scikit-learn (Pedregosa et al., 2011; Raschka et al., 2020), MNE (Gramfort et al., 2013), YASA (Vallat & Walker, 2021), and tsflex (Van Der Donckt et al., 2021). Features within *FeatLong* were mainly extracted using tsflex, as detailed in Appendix A.

Glass-box model hyperparameters:

- `NormIntSleep`-CatBoost: iterations = 5000, learning rate = 0.01, objective = MultiClass
- FeatShort-CatBoost: iterations = 10000, learning rate = 0.01, objective = MultiClass
- `NormIntSleep`-XGBoost: tree_method = gpu_hist, objective = multi:softprob, num_round = 30
- FeatShort-XGBoost: tree_method = gpu_hist, objective = multi:softprob, num_round = 200
- Decision Tree: criterion = entropy
- Logistic Regression: tol=1e-5, C=150.0, max_iter=4000, penalty=elasticnet, l1_ratio = 0.5

## C   VARIATION IN PERFORMANCE WITH DEPTH OF DECISION TREE

The performance comparison of decision trees of various depths are compared in Figure 4 and 5

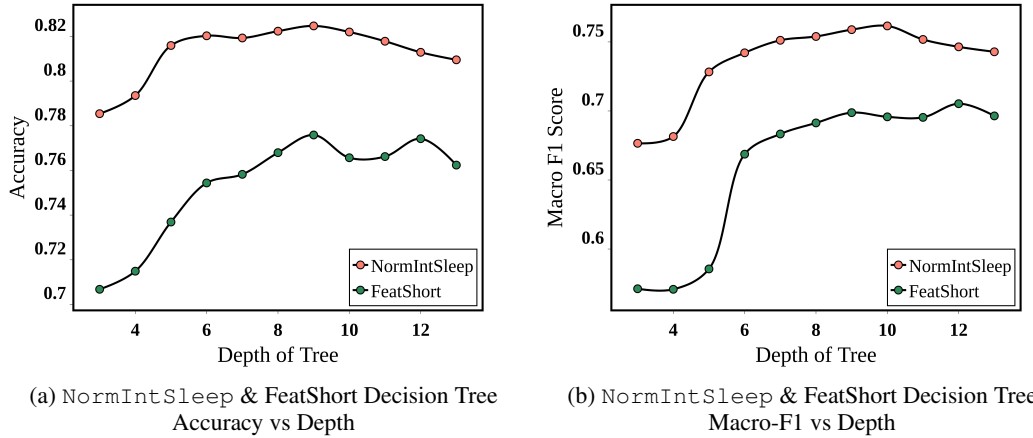

(a) `NormIntSleep` & FeatShort Decision Tree Accuracy vs Depth

(b) `NormIntSleep` & FeatShort Decision Tree Macro-F1 vs Depth

Figure 4: The performance variation with depth of tree in `NormIntSleep`-FeatShort Decision Tree

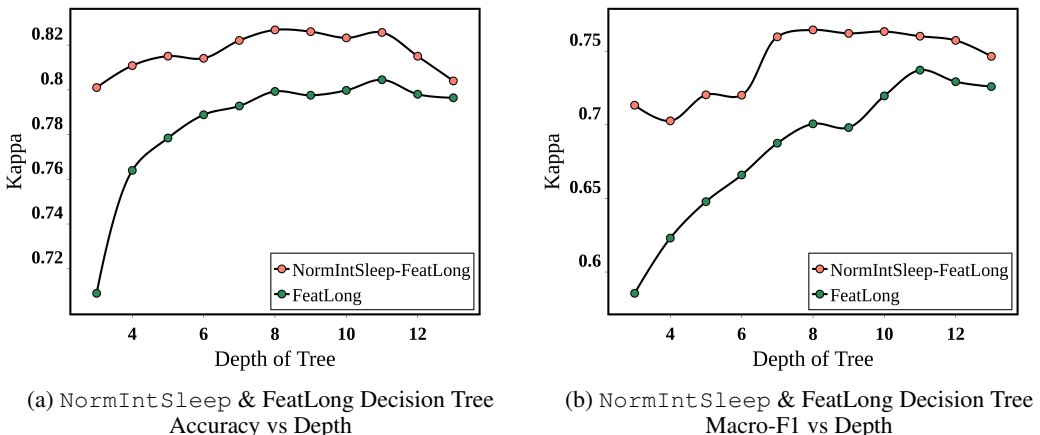

(a) `NormIntSleep` & FeatLong Decision Tree Accuracy vs Depth

(b) `NormIntSleep` & FeatLong Decision Tree Macro-F1 vs Depth

Figure 5: The performance variation with depth of tree in `NormIntSleep`-FeatLong Decision Tree

Figure 4 shows that even at a depth of 4 or 6, `NormIntSleep` reaches excellent performance metrics. FeatLong requires similar depth for good performance (Figure 5).

# D  ABLATION STUDY

The results of the ablation study on the two critical components of `NormIntSleep` are provided in Table 5 for the scaling schema and Table 4 for the choice of the projector learning algorithm. The tables show that the chosen approaches, Standardization and Least Squares, have comparable performance to the best methods.

Table 4: Effect of Projector Learning Schema

| Model | Scaling | Projector Learning | Accuracy | | F1 Score (Macro) | | Cohen's $\kappa$ | |
|---|---|---|---|---|---|---|---|---|
| | | | Physionet | ISRUC | Physionet | ISRUC | Physionet | ISRUC |
| NormIntSleep-XGBoost | Standardization | Cosine Similarity | 0.835 | 0.802 | 0.774 | 0.779 | 0.771 | 0.744 |
| FeatShort-XGBoost | Standardization | Cosine Similarity | 0.83 | 0.793 | 0.77 | 0.753 | 0.766 | 0.731 |
| NormIntSleep-CatBoost | Standardization | Cosine Similarity | 0.839 | 0.807 | 0.779 | 0.783 | 0.777 | 0.75 |
| FeatShort-CatBoost | Standardization | Cosine Similarity | 0.834 | 0.798 | 0.776 | 0.755 | 0.771 | 0.737 |
| NormIntSleep-LogisticRegression | Standardization | Cosine Similarity | 0.851 | 0.793 | 0.792 | 0.769 | 0.793 | 0.73 |
| FeatShort-LogisticRegression | Standardization | Cosine Similarity | 0.758 | 0.756 | 0.667 | 0.704 | 0.657 | 0.681 |
| NormIntSleep-XGBoost | Standardization | Linear Regression | 0.841 | 0.807 | 0.782 | 0.78 | 0.78 | 0.75 |
| FeatShort-XGBoost | Standardization | Linear Regression | 0.832 | 0.793 | 0.771 | 0.754 | 0.768 | 0.731 |
| NormIntSleep-CatBoost | Standardization | Linear Regression | 0.847 | 0.814 | 0.793 | 0.787 | 0.788 | 0.759 |
| FeatShort-CatBoost | Standardization | Linear Regression | 0.834 | 0.798 | 0.776 | 0.755 | 0.771 | 0.737 |
| NormIntSleep-LogisticRegression | Standardization | Linear Regression | 0.853 | 0.789 | 0.797 | 0.765 | 0.796 | 0.724 |
| FeatShort-LogisticRegression | Standardization | Linear Regression | 0.758 | 0.756 | 0.667 | 0.705 | 0.657 | 0.681 |
| NormIntSleep-XGBoost | Standardization | Ridge Regression | 0.841 | 0.803 | 0.782 | 0.776 | 0.78 | 0.744 |
| FeatShort-XGBoost | Standardization | Ridge Regression | 0.832 | 0.793 | 0.771 | 0.754 | 0.768 | 0.731 |
| NormIntSleep-CatBoost | Standardization | Ridge Regression | 0.847 | 0.814 | 0.793 | 0.786 | 0.789 | 0.758 |
| FeatShort-CatBoost | Standardization | Ridge Regression | 0.834 | 0.798 | 0.776 | 0.755 | 0.771 | 0.737 |
| NormIntSleep-LogisticRegression | Standardization | Ridge Regression | 0.853 | 0.788 | 0.797 | 0.763 | 0.796 | 0.723 |
| FeatShort-LogisticRegression | Standardization | Ridge Regression | 0.758 | 0.756 | 0.666 | 0.704 | 0.657 | 0.681 |
| NormIntSleep-XGBoost | Standardization | Least Squares | 0.841 | 0.807 | 0.782 | 0.78 | 0.78 | 0.75 |
| FeatShort-XGBoost | Standardization | Least Squares | 0.832 | 0.793 | 0.771 | 0.754 | 0.768 | 0.731 |
| NormIntSleep-CatBoost | Standardization | Least Squares | 0.847 | 0.814 | 0.793 | 0.787 | 0.788 | 0.759 |
| FeatShort-CatBoost | Standardization | Least Squares | 0.834 | 0.798 | 0.776 | 0.755 | 0.771 | 0.737 |
| NormIntSleep-LogisticRegression | Standardization | Least Squares | 0.853 | 0.788 | 0.797 | 0.764 | 0.796 | 0.723 |
| FeatShort-LogisticRegression | Standardization | Least Squares | 0.758 | 0.756 | 0.667 | 0.704 | 0.657 | 0.681 |

Table 5: Effect of Scaling Schema

| Model | Scaling | Projector Learning | Accuracy | | F1 Score (Macro) | | Cohen's $\kappa$ | |
|---|---|---|---|---|---|---|---|---|
| | | | Physionet | ISRUC | Physionet | ISRUC | Physionet | ISRUC |
| NormIntSleep-XGBoost | Max Absolute | Least Squares | 0.841 | 0.806 | 0.78 | 0.779 | 0.779 | 0.748 |
| FeatShort-XGBoost | Max Absolute | Least Squares | 0.833 | 0.792 | 0.772 | 0.752 | 0.769 | 0.729 |
| NormIntSleep-CatBoost | Max Absolute | Least Squares | 0.844 | 0.81 | 0.786 | 0.781 | 0.783 | 0.753 |
| FeatShort-CatBoost | Max Absolute | Least Squares | 0.834 | 0.797 | 0.775 | 0.755 | 0.771 | 0.736 |
| NormIntSleep-LogisticRegression | Max Absolute | Least Squares | 0.852 | 0.789 | 0.795 | 0.764 | 0.796 | 0.724 |
| FeatShort-LogisticRegression | Max Absolute | Least Squares | 0.751 | 0.741 | 0.653 | 0.684 | 0.646 | 0.66 |
| NormIntSleep-XGBoost | Normalization | Least Squares | 0.838 | 0.804 | 0.776 | 0.776 | 0.776 | 0.745 |
| FeatShort-XGBoost | Normalization | Least Squares | 0.827 | 0.791 | 0.765 | 0.747 | 0.762 | 0.728 |
| NormIntSleep-CatBoost | Normalization | Least Squares | 0.843 | 0.806 | 0.785 | 0.774 | 0.783 | 0.748 |
| FeatShort-CatBoost | Normalization | Least Squares | 0.831 | 0.799 | 0.769 | 0.753 | 0.766 | 0.738 |
| NormIntSleep-LogisticRegression | Normalization | Least Squares | 0.302 | 0.789 | 0.093 | 0.767 | 0 | 0.725 |
| FeatShort-LogisticRegression | Normalization | Least Squares | 0.694 | 0.754 | 0.578 | 0.691 | 0.562 | 0.677 |
| NormIntSleep-XGBoost | Standardization | Least Squares | 0.841 | 0.807 | 0.782 | 0.78 | 0.78 | 0.75 |
| FeatShort-XGBoost | Standardization | Least Squares | 0.832 | 0.793 | 0.771 | 0.754 | 0.768 | 0.731 |
| NormIntSleep-CatBoost | Standardization | Least Squares | 0.847 | 0.814 | 0.793 | 0.787 | 0.788 | 0.759 |
| FeatShort-CatBoost | Standardization | Least Squares | 0.834 | 0.798 | 0.776 | 0.755 | 0.771 | 0.737 |
| NormIntSleep-LogisticRegression | Standardization | Least Squares | 0.853 | 0.788 | 0.797 | 0.764 | 0.796 | 0.723 |
| FeatShort-LogisticRegression | Standardization | Least Squares | 0.758 | 0.756 | 0.667 | 0.704 | 0.657 | 0.681 |
| NormIntSleep-XGBoost | Quantile | Least Squares | 0.844 | 0.807 | 0.786 | 0.778 | 0.784 | 0.749 |
| FeatShort-XGBoost | Quantile | Least Squares | 0.833 | 0.793 | 0.773 | 0.753 | 0.77 | 0.731 |
| NormIntSleep-CatBoost | Quantile | Least Squares | 0.847 | 0.809 | 0.791 | 0.781 | 0.788 | 0.752 |
| FeatShort-CatBoost | Quantile | Least Squares | 0.834 | 0.797 | 0.776 | 0.755 | 0.771 | 0.736 |
| NormIntSleep-LogisticRegression | Quantile | Least Squares | 0.854 | 0.789 | 0.798 | 0.766 | 0.798 | 0.724 |
| FeatShort-LogisticRegression | Quantile | Least Squares | 0.793 | 0.781 | 0.72 | 0.734 | 0.712 | 0.714 |
| NormIntSleep-XGBoost | Robust | Least Squares | 0.842 | 0.807 | 0.78 | 0.781 | 0.78 | 0.75 |
| FeatShort-XGBoost | Robust | Least Squares | 0.833 | 0.795 | 0.772 | 0.755 | 0.769 | 0.733 |
| NormIntSleep-CatBoost | Robust | Least Squares | 0.842 | 0.807 | 0.784 | 0.778 | 0.781 | 0.749 |
| FeatShort-CatBoost | Robust | Least Squares | 0.834 | 0.798 | 0.775 | 0.755 | 0.771 | 0.737 |
| NormIntSleep-LogisticRegression | Robust | Least Squares | 0.302 | 0.792 | 0.093 | 0.769 | 0 | 0.728 |
| FeatShort-LogisticRegression | Robust | Least Squares | 0.758 | 0.757 | 0.667 | 0.705 | 0.657 | 0.682 |
| NormIntSleep-XGBoost | None | Least Squares | 0.84 | 0.806 | 0.781 | 0.779 | 0.778 | 0.748 |
| FeatShort-XGBoost | None | Least Squares | 0.833 | 0.792 | 0.772 | 0.752 | 0.769 | 0.729 |
| NormIntSleep-CatBoost | None | Least Squares | 0.844 | 0.81 | 0.786 | 0.781 | 0.783 | 0.753 |
| FeatShort-CatBoost | None | Least Squares | 0.834 | 0.797 | 0.776 | 0.755 | 0.771 | 0.736 |
| NormIntSleep-LogisticRegression | None | Least Squares | 0.302 | 0.789 | 0.093 | 0.764 | 0 | 0.724 |
| FeatShort-LogisticRegression | None | Least Squares | 0.722 | 0.755 | 0.607 | 0.702 | 0.602 | 0.679 |

## E   NᴏʀᴍIɴᴛSʟᴇᴇᴘ-XGBᴏᴏsᴛ Fᴇᴀᴛᴜʀᴇ Iᴍᴘᴏʀᴛᴀɴᴄᴇ

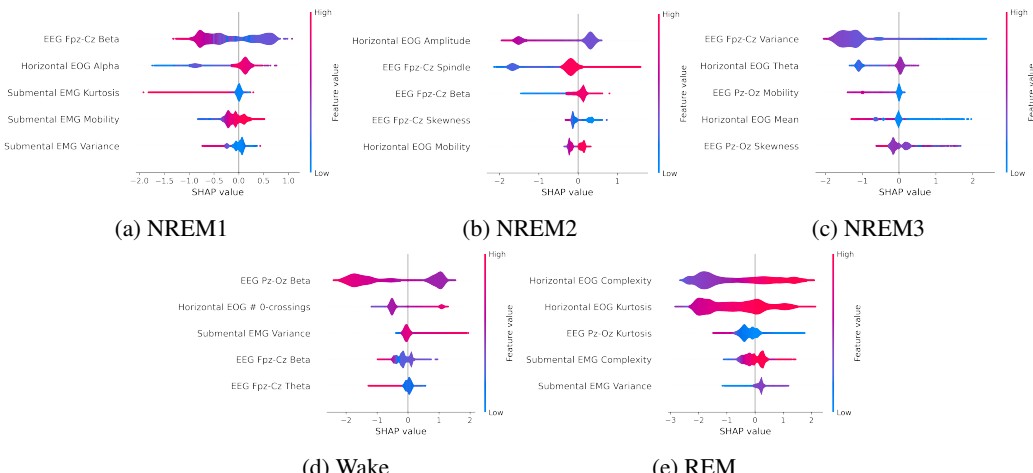

Figure 6: Five most important features of `NormIntSleep`-XGBoost during the classification of (a) NREM1, (b) NREM2, (c) NREM3, (d) Wake, and (e) REM

# F  BASELINE DEEP LEARNING MODEL ARCHITECTURE

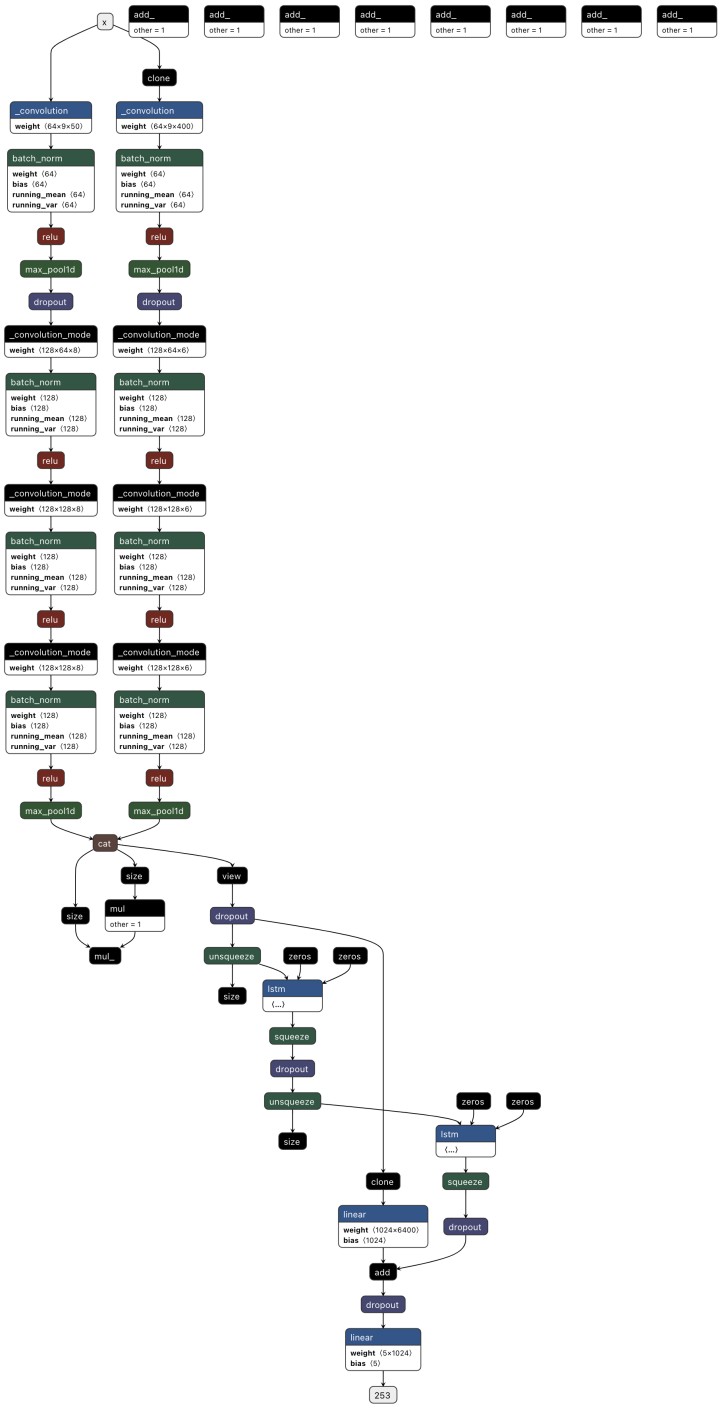

Figure 7: Multi-channel DeepSleepNet for the ISRUC dataset

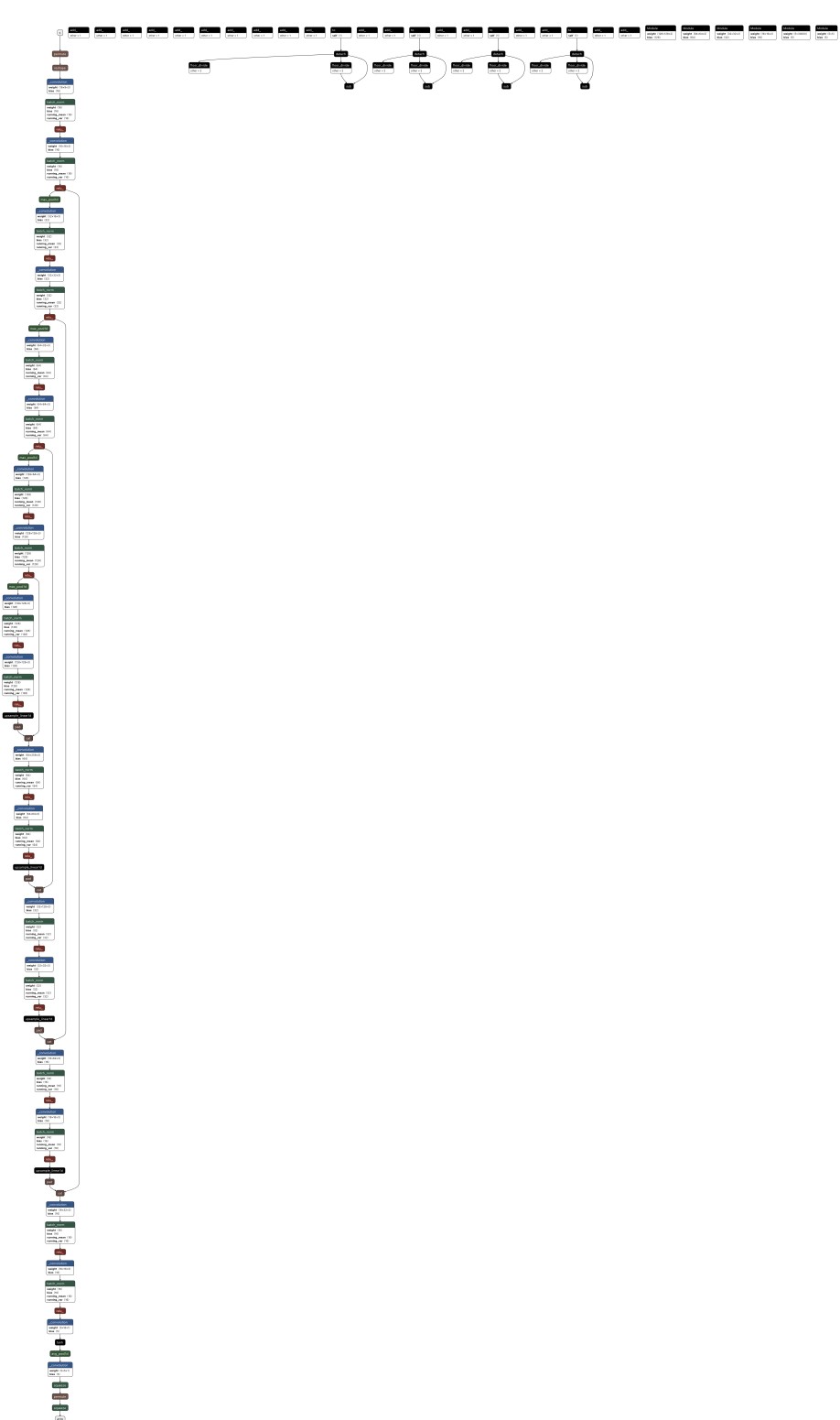

Figure 8: Multi-channel U-Time for the ISRUC dataset

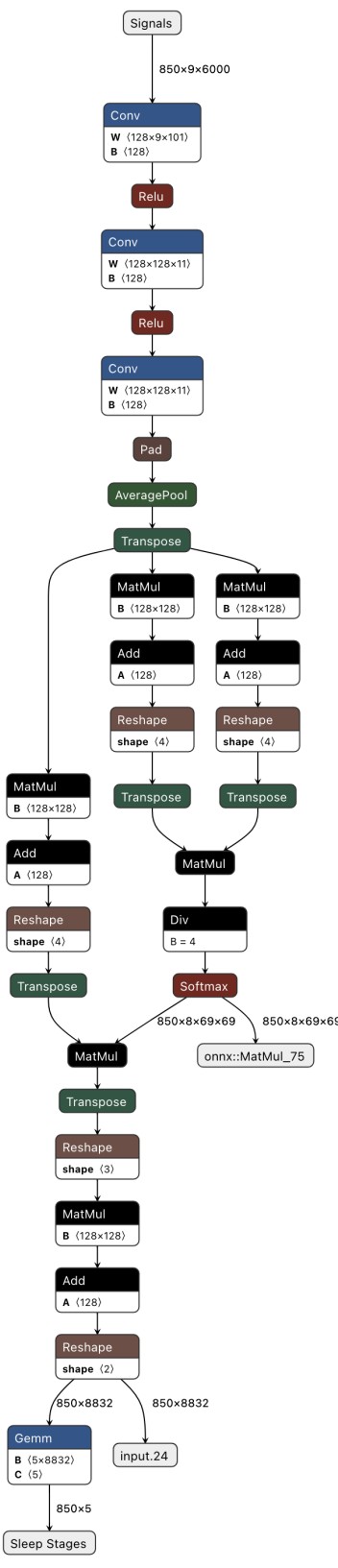

Figure 9: AttentionNet for the ISRUC dataset

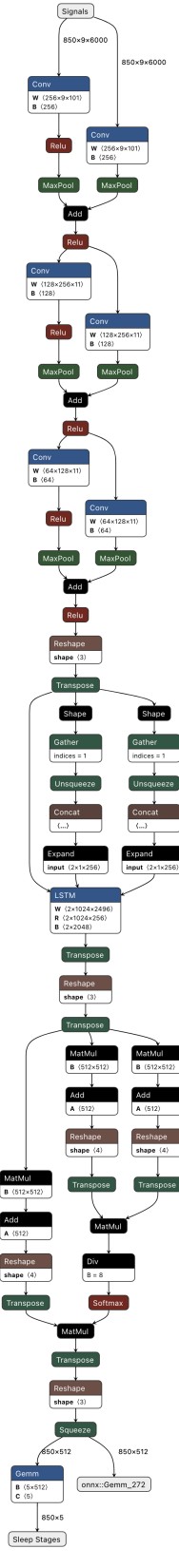

Figure 10: RCNN-MHA for the ISRUC dataset

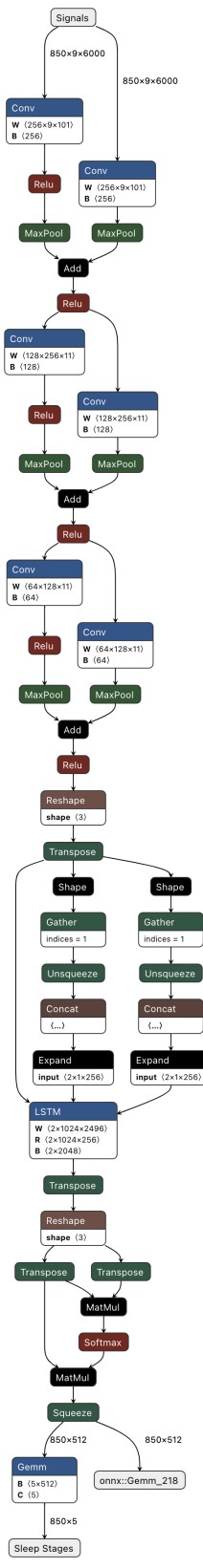

Figure 11: RCNN-SDPA for the ISRUC dataset

## G  SLEEP STAGE PERFORMANCE

Table 6: F1 scores for each sleep stage in the Physionet dataset

| | Model | W | N1 | N2 | N3 | REM |
|---|---|---|---|---|---|---|
| **Interpretable Methods** | FeatLong (Van Der Donckt et al., 2022)-XGBoost | 0.941 | 0.541 | 0.881 | 0.822 | 0.867 |
| | **FeatLong (Van Der Donckt et al., 2022)-CatBoost** | 0.938 | 0.530 | 0.884 | 0.836 | 0.866 |
| | FeatLong (Van Der Donckt et al., 2022)-LogisticRegression | 0.930 | 0.511 | 0.882 | 0.833 | 0.849 |
| | FeatShort-XGBoost | 0.916 | 0.437 | 0.859 | 0.828 | 0.801 |
| | NormIntSleep-XGBoost | 0.922 | 0.495 | 0.870 | 0.824 | 0.826 |
| | FeatShort-CatBoost | 0.918 | 0.445 | 0.862 | 0.846 | 0.808 |
| | **NormIntSleep-CatBoost** | 0.921 | 0.515 | 0.875 | 0.826 | 0.827 |
| | NormIntSleep-LogisticRegression | 0.932 | 0.496 | 0.874 | 0.846 | 0.834 |
| | FeatShort-Decision Tree (Depth 7) | 0.838 | 0.293 | 0.815 | 0.790 | 0.681 |
| | NormIntSleep-Decision Tree (Depth 7) | 0.898 | 0.396 | 0.858 | 0.816 | 0.788 |
| **Deep Learning** | U-Time (Perslev et al., 2019) | 0.909 | 0.385 | 0.831 | 0.763 | 0.776 |
| | DeepSleepNet (Supratak et al., 2017) | 0.920 | 0.495 | 0.873 | 0.832 | 0.820 |
| | CNN (Al-Hussaini et al., 2019) | 0.938 | 0.491 | 0.871 | 0.843 | 0.839 |
| | AttnSleep (Eldele et al., 2021) | 0.921 | 0.438 | 0.871 | 0.848 | 0.783 |
| | TinySleepNet (Supratak & Guo, 2020) | 0.906 | 0.321 | 0.864 | 0.838 | 0.724 |
| | **U-Time (multi-channel)** | 0.947 | 0.579 | 0.882 | 0.817 | 0.885 |
| | DeepSleepNet (multi-channel) | 0.941 | 0.560 | 0.882 | 0.839 | 0.879 |
| | AttentionNet | 0.932 | 0.447 | 0.863 | 0.838 | 0.824 |
| | RCNN (DNN in NormIntSleep) | 0.933 | 0.561 | 0.888 | 0.819 | 0.883 |
| | RCNN-MHA | 0.937 | 0.551 | 0.874 | 0.821 | 0.872 |
| | RCNN-SDPA | 0.943 | 0.573 | 0.880 | 0.830 | 0.888 |

Table 7: F1 scores for each sleep stage in the ISRUC dataset

| | Model | W | N1 | N2 | N3 | REM |
|---|---|---|---|---|---|---|
| **Interpretable Methods** | FeatLong (Van Der Donckt et al., 2022)-XGBoost | 0.864 | 0.431 | 0.815 | 0.900 | 0.867 |
| | **FeatLong (Van Der Donckt et al., 2022)-CatBoost** | 0.874 | 0.419 | 0.815 | 0.902 | 0.863 |
| | FeatLong (Van Der Donckt et al., 2022)-LogisticRegression | 0.889 | 0.412 | 0.790 | 0.877 | 0.841 |
| | FeatShort-XGBoost | 0.899 | 0.401 | 0.781 | 0.873 | 0.804 |
| | NormIntSleep-XGBoost | 0.893 | 0.499 | 0.796 | 0.869 | 0.857 |
| | FeatShort-CatBoost | 0.902 | 0.397 | 0.794 | 0.878 | 0.805 |
| | **NormIntSleep-CatBoost** | 0.895 | 0.507 | 0.799 | 0.867 | 0.861 |
| | NormIntSleep-LogisticRegression | 0.891 | 0.501 | 0.762 | 0.841 | 0.832 |
| | FeatShort-Decision Tree (Depth 7) | 0.838 | 0.293 | 0.815 | 0.790 | 0.681 |
| | NormIntSleep-Decision Tree (Depth 7) | 0.872 | 0.462 | 0.783 | 0.833 | 0.852 |
| **Deep Learning** | U-Time (Perslev et al., 2019) | 0.890 | 0.451 | 0.764 | 0.883 | 0.785 |
| | DeepSleepNet (Supratak et al., 2017) | 0.916 | 0.500 | 0.796 | 0.881 | 0.800 |
| | CNN (Al-Hussaini et al., 2019) | 0.904 | 0.459 | 0.794 | 0.878 | 0.855 |
| | AttnSleep (Eldele et al., 2021) | 0.914 | 0.418 | 0.799 | 0.887 | 0.783 |
| | TinySleepNet (Supratak & Guo, 2020) | 0.911 | 0.372 | 0.781 | 0.884 | 0.740 |
| | **U-Time (multi-channel)** | 0.934 | 0.576 | 0.827 | 0.897 | 0.893 |
| | DeepSleepNet (multi-channel) | 0.929 | 0.566 | 0.795 | 0.835 | 0.887 |
| | AttentionNet | 0.920 | 0.451 | 0.779 | 0.876 | 0.846 |
| | RCNN (DNN in NormIntSleep) | 0.923 | 0.581 | 0.818 | 0.890 | 0.901 |
| | RCNN-MHA | 0.904 | 0.505 | 0.790 | 0.864 | 0.879 |
| | RCNN-SDPA | 0.869 | 0.461 | 0.733 | 0.809 | 0.834 |

# H Interpretation of FeatLong-Decision Tree

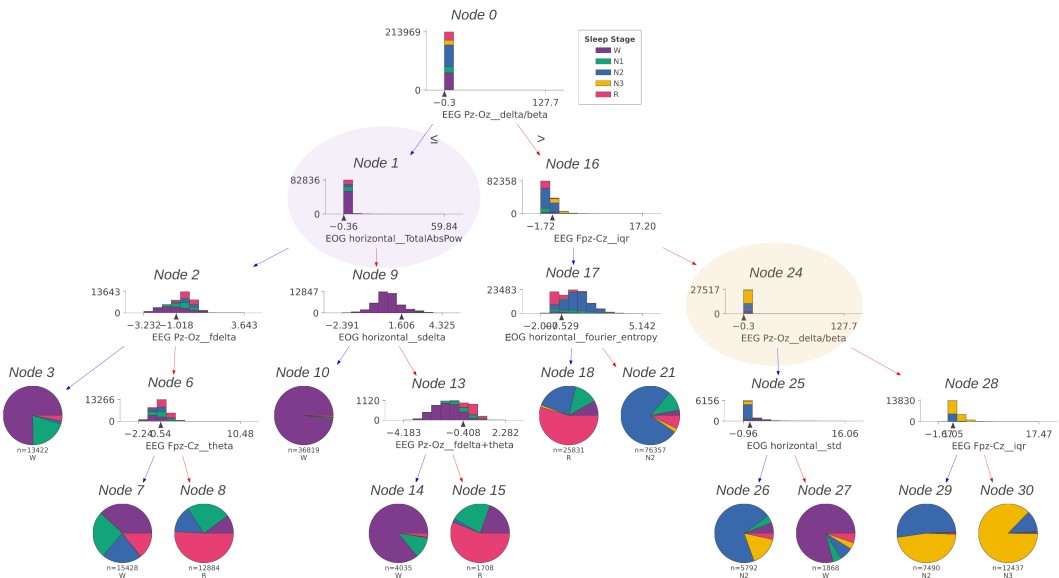

Figure 12: The *FeatLong* (Van Der Donckt et al., 2022)-DecisionTree (Depth 4) features just 2 nodes (Nodes 1 and 24) aligned with clinical guidelines, yielding an $Alignment_{DT}$ score of 0.22. Nodes 9 and 13 are excluded from the $Alignment_{DT}$ calculation since they predominantly represent Wake ($> 95\%$). In contrast, the `NormIntSleep`-DecisionTree (Depth 4) achieves a perfect $Alignment_{DT}$ score of 1.0, with all its nodes in alignment.

# I Reproducibility

Table 8: Performance metrics obtained from repeated experiments with disjoint test subjects on both datasets showing average and 95% confidence interval.

| | PhysioNet Accuracy | ISRUC Accuracy | PhysioNet F1 Score (Macro) | ISRUC F1 Score (Macro) | PhysioNet Kappa | ISRUC Kappa |
|---|---|---|---|---|---|---|
| RCNN | 0.846±0.013 | 0.826±0.017 | 0.804±0.017 | 0.808±0.017 | 0.788±0.019 | 0.774±0.024 |
| U-Time | 0.818±0.014 | 0.835±0.010 | 0.768±0.015 | 0.815±0.010 | 0.750±0.021 | 0.785±0.015 |
| NormIntSleep-CatBoost | 0.805±0.023 | 0.801±0.011 | 0.753±0.020 | 0.778±0.013 | 0.730±0.031 | 0.739±0.016 |
| NormIntSleep-XGBoost | 0.796±0.022 | 0.799±0.009 | 0.742±0.021 | 0.775±0.009 | 0.717±0.031 | 0.737±0.013 |
| NormIntSleep-Logistic Regression | 0.802±0.023 | 0.799±0.001 | 0.747±0.023 | 0.778±0.008 | 0.725±0.033 | 0.736±0.001 |
| FeatLong-CatBoost | 0.849±0.014 | 0.814±0.003 | 0.803±0.011 | 0.791±0.013 | 0.792±0.019 | 0.756±0.003 |
| FeatLong-XGBoost | 0.850±0.012 | 0.810±0.002 | 0.803±0.010 | 0.788±0.009 | 0.792±0.017 | 0.751±0.003 |
| FeatLong-Logistic Regression | 0.832±0.025 | 0.794±0.015 | 0.780±0.023 | 0.767±0.010 | 0.768±0.034 | 0.731±0.019 |

# J Results by Demography for ISRUC Dataset

Table 9: Performance Variation with Age showing average and 95% confidence interval.

| Age | Accuracy | F1 Score (Macro) | Kappa |
|---|---|---|---|
| <35 | $0.855 \pm 0.010$ | $0.822 \pm 0.023$ | $0.807 \pm 0.016$ |
| 35-60 | $0.726 \pm 0.184$ | $0.702 \pm 0.182$ | $0.644 \pm 0.223$ |
| 61< | $0.837 \pm 0.038$ | $0.806 \pm 0.043$ | $0.785 \pm 0.046$ |

These findings suggest our model's consistent robustness across different genders and age brackets, albeit with minor deviations for male subjects and those aged between 35-60. Unfortunately, the PhysioNet dataset does not provide demography-specific data tied to individual files, limiting our analysis on that front.

Table 10: Performance Variation with Sex showing average and 95% confidence interval.

| Sex | Accuracy | F1 Score (Macro) | Kappa |
|---|---|---|---|
| Male | $0.782 \pm 0.123$ | $0.755 \pm 0.122$ | $0.717 \pm 0.150$ |
| Female | $0.84 \pm 0.032$ | $0.808 \pm 0.035$ | $0.786 \pm 0.046$ |

Table 11: Annotation Distribution with Age.

| Age | W | N1 | N2 | N3 | R |
|---|---|---|---|---|---|
| $0 - 35$ | 0.216 | 0.085 | 0.343 | 0.227 | 0.129 |
| $35 - 60$ | 0.173 | 0.134 | 0.313 | 0.232 | 0.148 |
| $61 <$ | 0.306 | 0.137 | 0.266 | 0.145 | 0.146 |

Table 12: Annotation Distribution with Sex.

| Sex | W | N1 | N2 | N3 | R |
|---|---|---|---|---|---|
| Male | 0.243 | 0.112 | 0.292 | 0.231 | 0.123 |
| Female | 0.224 | 0.119 | 0.327 | 0.174 | 0.155 |

## K  ADOPTION PROCESS IN OTHER DOMAINS

1. Identify and list features based on domain knowledge similar to Section 3.3. This requires domain expertise.

2. Determine methods to extract the selected features from the training data. This step equires an extensive research of prior work on feature extraction.

3. Normalize the extracted features.

4. Design an end-to-end deep-learning model that has a fully-connected final layer for classification.

5. Remove the final fully-connected layer connecting the neural network's last embeddings to output nodes.

6. Retrieve embeddings from the training data.

7. Use the embeddings from Step 6 and the normalized domain-grounded feature space defined in Step 3 to learn a linear projector matrix to transform embeddings into the domain-grounded feature space.

8. Apply the linear projector on the embeddings to obtain interpretable representations for training data.

9. Finally, utilize these interpretable representations to train a glass-box model.

10. For inference, first the interpretable representations are obtained using the linear projector to project the embedding output from the trained deep learning model to the interpretable feature space. Then these features are used as input to the trained glass-box model.

11. Use a decision tree as the glass-box model to evaluate domain-grounded interpretability using $Alignment_{DT}$.

These processes are further detailed in Figure 1 as well as Algorithms 1 and 2.

