# OpenReview forum: "Optimizing the trade-off between utility and performance in interpretable sleep classification"
_ICLR.cc/2024/Conference — Submitted to ICLR 2024_

### Official Review · Reviewer_ppPz · 2023-10-21

**Soundness:** 3 good
**Presentation:** 2 fair
**Contribution:** 2 fair
**Rating:** 3
**Confidence:** 4

**Summary:**

This paper proposes an interpretable method for sleep staging, combining handcrafted features and deep neural networks. The authors also propose a metric to quantify interpretability of the model. Experiments suggest that the proposed model provides significantly better interpretability than a feature-based model.

**Strengths:**

1. This paper addresses an important application in ML for healthcare: providing interpretability to sleep staging algorithms while maintaining a high accuracy.
2. Extensive experiments were performed on 2 public datasets and various baselines.

**Weaknesses:**

1. Clarity can be improved, particularly in Methods section. For example, in section 3.1, the authors introduce pre-training and point to Algorithm 1 and Figure 1a, but abruptly switch to introducing a DNN trained end-to-end on sleep staging. Also, notations are overloaded, which reduces the readability of the manuscript.
2. There is no details about hypnogram except for Figure 1 and the second last paragraph of page 4.
3. The proposed metric is only applicable for decision trees, which hinders its utility to other glass-box algorithms.
4. Technical contribution is limited. The DNN architecture (CNN+LSTM) are commonly used in sleep staging literature. FeatShort features are also adapted from prior work (Al-Hussaini & Mitchell, 2022).

**Questions:**

1. In pre-training stage, how is cross-entropy loss for DNN combined with regression loss? A simple average? Please clarify in the manuscript.
2. How is Alignment metric calculated? Did a clinician manually look at the nodes and calculate the metric? Any consensus among multiple clinicians?
3. Please provide details about hypnogram.
4. Please acknowledge the limitation that the metric can only be used for decision trees.

---

### Official Review · Reviewer_uEgX · 2023-10-31

**Soundness:** 1 poor
**Presentation:** 1 poor
**Contribution:** 2 fair
**Rating:** 3
**Confidence:** 2

**Summary:**

The authors of this paper propose an interpretable method for sleep stage classification aiming to close the gap on state-of-the-art non-interpretable models for sleep stage classification. Their method has two stages. First, train a deep neural network (DNN) to classify sleep stages jointly with a linear projector which projects the DNN embeddings (penultimate layer) into a pre-defined feature space. These features are hand-extracted and clinically motivated for sleep stage definition. In the second stage, an interpretable model (boosted trees here), is trained on the projection from the DNN embeddings onto the clinical features space. Because embedding's projections are aligned with clinical features, the authors propose to interpret them as so making their model directly interpretable w.r.t to clinical features importance.

**Strengths:**

This work tackles a relevant problem, the need for clinically interpretable models for the acceptance of machine learning in healthcare. Also, the effort in terms of experiment is significant. Below is a detailed assessment of the strength of the paper:

### Relevance of the task

As noted by the authors, interpretability and explainability are key to the acceptance of ML by clinicians. The authors do a great job of motivating the problem and highlighting the gap between interpretable models based on clinical features and black-box or non-clinically relevant models.


### Experiments
The authors made a great effort to consider two datasets and many baselines. In particular, they made the effort to adapt possible baselines to their framework to have a more fair comparison. The results in terms of performance clearly show what the authors wanted to highlight, a trade-off between the clinical feature-based model and DL. Given it is valid, (see weakness below) the interpretability section is a very nice addition.

**Weaknesses:**

### Questioning concerning the interpretability of the model
The authors learn a linear projector from DNN embeddings onto (a scaled version of) a set of features and they use these projections, as if they were the features themselves, to train a glass-box classifier. In that way, they hope to achieve better performance than training directly on the features.

For me, this is problematic from an interpretability point of view. Indeed, if the linear model can perfectly regress the features, then training on the projection or the features is strictly equivalent. In that case, interpreting the projections as the features themselves is fine, as they are identical. However, the performance is also identical, thus less good than the black-box models. On the other hand, as the regression error increases between features and projections, the downstream performance might diverge (in a positive way) as well. However, the more projections and features diverge, the less one can assume to interpret them as if they were the features themselves.

Thus, I believe the method to be ill-posed, as improved performance can only be achieved at the cost of guarantees in terms of interpretability.


### Clarity
The second main weakness of this work is the overall clarity in my opinion. After carefully reading both the main body and supplementary, I found the paper to still be quite hard to understand. Here are some examples:
- The ordering of sections, hurts the reading with typically the second section being a presentation of the data, before even knowing about the task or method of interest.
- The choice of the term "epoch" to refer to a 30s segment of sleep recording which is highly confusing with the term referring to a full pass on a dataset when training a neural network.
- The definition of a new metric where terms are not properly defined. What does alignment between nodes and domain knowledge mean?
- The authors refer to "glass-box models" throughout the paper without defining it. I understand it's a common terminology in interpretability but it should be defined.

### Lack of related work
Linking to the lack of clarity, I believe the absence of related work is both a limitation for putting the work in the context of the field and also for the clarity itself.
Typically, clear links exist in this work with the ideas of partial concept bottleneck models, which aim to create a similar trade-off between performance and maintaining concept-level interpretability.

### Other issues
- No confidence interval for the experiments
- The authors did not always bold the best method in their results

## Conclusion
If this work tackles a highly relevant problem in the field of clinical ML, closing performance on black-box model while preserving interpretable models, the limitations of the proposed method in terms of guarantee for interoperability as well as the lack of clarity and related work push me to recommend rejection. Because I had a hard time understanding parts of the manuscript, some of my concerns may be in fact caused by my lack of understanding and not justified, hence my lower confidence score.

**Questions:**

The authors state "Model hyperparameters were optimized based on the training data", what does that mean in practice? Did they use a cross-validation?

---

### Official Review · Reviewer_coUd · 2023-11-01

**Soundness:** 2 fair
**Presentation:** 3 good
**Contribution:** 2 fair
**Rating:** 3
**Confidence:** 4

**Summary:**

This paper proposes a framework, NormIntSleep, which utilizes a pre-trained deep learning model with high sleep stage classification performance for domain-specific feature interpretation. It transforms the learned embeddings to domain-specific interpretable feature space with a linear projector trained with clinically relevant features designed based on domain-specific knowledge. The authors state this framework outperforms other approaches that aim for clinically relevant interpretations, and also propose a new metric, AlignmentDT, that measures alignment between model and domain-specific knowledge in a decision tree.

**Strengths:**

- The paper deals with an important topic in sleep stage classification and healthcare where explanation of features derived from domain-specific knowledge is crucial
- The Interpretable Decision Tree in section 5.1 and figure 2 introduces a practical way of sleep stage explanation
- The paper is easy to follow

**Weaknesses:**

- The conducted experiments are not sufficient to convince the reader that the interpretations from NormIntSleep is entirely meaningful: The sleep clinician made comments on the output of NormIntSleep-Decision Tree method, which is not the best performing model indicated in Table 3. Also, only SHAP method is used for feature importance. Comparison with other feature-based interpretable methods can be a starting point for addressing the soundness aspect of this work.
- The approach does not seem to be generalizable to other domains, especially when the best performing glass-box model is not the decision tree, because AlignmentDT cannot be used
- Some details are missing in the paper (for example in section 3.2, how the hypnogram is used alongside the interpretable representations)

**Questions:**

Please see the weaknesses.

---

### Official Review · Reviewer_ommx · 2023-11-09

**Soundness:** 3 good
**Presentation:** 3 good
**Contribution:** 3 good
**Rating:** 5
**Confidence:** 3

**Summary:**

This paper tackles the problem of training accurate yet interpretable sleep-stage decoders. To this end, the authors introduce NormIntSleep, a framework in which they (1) train a deep neural network A to classify sleep stages and then (2) train a linear model which maps the latent representation learnt by A to different feature space which is interpretable by design. Finally, they feed these interpretable features to a glass-box model predicting sleep stages. Moreover, the authors design a new metric to assess how well a glass-box model's decision nodes are consistent with pre-established domain knowledge.

They show that their approach is competitive with deep, non-interpretable models, while relying on feature spaces and decision algorithms whose mechanism are grounded in clinically established knowledge and protocols.

**Strengths:**

**Significance / originality** - I cannot really tell if this piece of work is original or not, but I agree with the authors that model interpretability is important for clinical applications. In my opinion, results reported in this paper are not paradigm-shifting, but I deem this is an interesting and valuable line of research, and that it is important to publicly report results comparable to that present in this piece of work

**Quality / Clarity** - In my opinion, this paper is generally well written and clear. The problem is well motivated, and a high number of cases were tested

**Weaknesses:**

In my opinion, the main weaknesses of this piece of work are:
- it is not clear why models trained on predicted features of a given feature space should perform better than models trained on ground-truth features of this same space
- it is not clear to me if one can make a fair comparison between models fitted on NormIntSleep and FeatLong respectively using the alignment metric. Indeed, it feels to me that, by construction, NormIntSleep features are similar to those used in the alignment metric, when FeatLong features are not. I might not understand what the alignment metric actually measures, which is also a problem in this paper.
- code to reproduce presented experiments is missing

Here is a more precise list of comments, suggestions and quick questions:
- in my opinion, Figure 1 could be clearer. In particular, I think panel A should help the reader understand that there are multiple steps represented (step 1: fit DNN, step 2: extract and normalise domain features, step 3: fit linear mapping between latent representation from DNN and domain features). Moreover, I find it misleading to use the boxes "embeddings" and "normalization" as inputs of the box "optimization". Finally, you could already add notations at this stage (for $T$, $F$, $E$, etc). Generally, it is not clear to me what the colour code used for boxes represents.
- although I find this paper generally well written, the overall layout feels a bit dense. In my opinion, Algorithm 1 and 2 are mildly informative and take a lot of space: maybe they could be moved to the appendix. I have a similar feeling about Fig1.c
	- As a side note, in algorithm 2 line 9, what is $R'$ compared to $R$? Similar question for $T'$ in algorithm 1
- equation 1: to me, a clear definition of what it means for a node to be "aligned with clinical domain knowledge" is still needed at this stage, otherwise the overall definition of the alignment metric feels very circular. In particular, do you use the threshold value to determine if a node is aligned? Moreover, it feels to me that this notation should specify what nodes are compared to (in this case, I guess it would be the AASM Manual, but details feel hazy to me)
- results are reported without standard error to the mean, so in my opinion it is hard to assess which approach is performing the best
- Table 3: I think this table would be easier to read if names of interpretable methods were split in two columns (features and glass-box model)

**Questions:**

I feel uncertain about some important aspects of this paper, and will probably update my rating based on responses to the following questions:
- in table 4: for projectors fitted using ridge regression, how did you choose the value of the regularization parameter?
- regarding the ablation study, how did you choose the final learning and scaling schemas?
	- in table 4: to me, cosine similarity for NormIntSleep-LogisticRegression seems comparable or marginally better than that of the other best combinations, and in particular NormIntSleep-LogReg Standardization Least Squares
	- in table5: similar observation with Quantile NormIntSleep-LogReg.
	- moreover, it's unclear to me what difference you make between Least Square and Linear Regression (the `LinearRegression` model of `sklearn`, which I assume you used, solves an Ordinary Least Squares linear regression problem)
- I am very intrigued by the fact that a glass-box model trained on ground-truth FeatShort would be out-performed by a class-box model trained on a prediction of FeatShort (i.e. `NormIntSleep`). Can you comment on why you think this is possible? My concern is that, if the linear projector had perfect accuracy when reconstructing features, there should not be any difference of sleep-stage decoding performance between models trained on ground-truth features and predicted features. Could it be that your framework somewhat regularises features from FeatShort in a way that makes them more prone to fitting a glass-box algorithm?
	- As a follow-up on this question, I would be very interested to see how a glass-box model trained on ground-truth FeatShort performs on predicted FeatShort (i.e. `NormIntSleep`) and vice-versa. Could you please include these results to better understand what is going on?
	- Moreover,  FeatShort-LogisticRegression seems to be missing from Table3, could you please add it?
- in Figure2: I am wondering what the same figure for a model trained on ground-truth FeatShort features would look like. It feels to me that $\text{Alignment}_{DT}$ should be very high for this model as well
- as stated in the "weaknesses" section, I have trouble understanding the $\text{Alignment}_{DT}$ metric ; however, I wonder if it is fair to compare `NormIntSleep` to FeatLong through this metric, given that `NormIntSleep` was built to have features similar to that of what the alignment metric expects. This feels like a form of double-dipping to me. Shouldn't you try to extract a subset of features from FeatLong, train your linear projector on this subset, and then compare alignment values to AASM?

---

### Meta-Review · Area_Chair_CjMj · 2023-12-07

**Metareview:**

This submission sparked some interest from reviewers. The topic was seen as interesting. However, it was not clear that the submission meets the bar for ICLR. Indeed, the reviewers were not fully convinced by the experiments: the method leads to a loss of prediction performance and with regards to interpretation, there is little comparison with existing interpretation methods. In addition, the approach seems tied to decision trees as an interpretable base model.

The reviewers also raised some subttle questions on whether the interpretability approach would not hit fundamental challenge due to the back and forth between simple and complex models. Unfortunately, these questions did not lead to discussion.

**Justification For Why Not Higher Score:**

The contribution is not very solid: both the theory and the experiments are not very solid.

**Justification For Why Not Lower Score:**

N/A

---

### Decision · Program_Chairs · 2024-01-16

Reject